# Outcomes of Patients Receiving a Kidney Transplant or Remaining on the Transplant Waiting List at the Epicentre of the COVID-19 Pandemic in Europe: An Observational Comparative Study

**DOI:** 10.3390/pathogens11101144

**Published:** 2022-10-03

**Authors:** Marta Perego, Samuele Iesari, Maria Teresa Gandolfo, Carlo Alfieri, Serena Delbue, Roberto Cacciola, Mariano Ferraresso, Evaldo Favi

**Affiliations:** 1Kidney Transplantation, Fondazione IRCCS Ca’ Granda Ospedale Maggiore Policlinico, 20122 Milan, Italy; 2Pôle de Chirurgie Expérimentale et Transplantation, Institut de Recherche Expérimentale et Clinique, Université Catholique de Louvain, 1348 Brussels, Belgium; 3Nephrology, Dialysis and Transplantation, Fondazione IRCCS Ca’ Granda Ospedale Maggiore Policlinico, 20122 Milan, Italy; 4Department of Clinical Sciences and Community Health, Università degli Studi di Milano, 20122 Milan, Italy; 5Department of Biomedical, Surgical and Dental Sciences, Università degli Studi di Milano, 20122 Milan, Italy; 6Surgery, King Salman Armed Forces Hospital, Tabuk 47512, Saudi Arabia; 7HPB Surgery and Transplantation, Fondazione PTV, 00133 Rome, Italy

**Keywords:** SARS-CoV-2, COVID-19, coronavirus, pandemic, kidney transplantation, chronic kidney disease, dialysis, outcomes

## Abstract

Since the declaration of the COVID-19 pandemic, the number of kidney transplants (KT) performed worldwide has plummeted. Besides the generalised healthcare crisis, this unprecedented drop has multiple explanations such as the risk of viral transmission through the allograft, the perceived increase in SARS-CoV-2-related morbidity and mortality in immunocompromised hosts, and the virtual “safety” of dialysis while awaiting effective antiviral prophylaxis or treatment. Our institution, operating at the epicentre of the COVID-19 pandemic in Italy, has continued the KT programme without pre-set limitations. In this single-centre retrospective observational study with one-year follow-up, we assessed the outcomes of patients who had undergone KT (KTR) or remained on the transplant waiting list (TWL), before (Pre-COV) or during (COV) the pandemic. The main demographic and clinical characteristics of the patients on the TWL or receiving a KT were very similar in the two periods. The pandemic did not affect post-transplant recipient and allograft loss rates. On the contrary, there was a trend toward higher mortality among COV-TWL patients compared to Pre-COV-TWL subjects. Such a discrepancy was primarily due to SARS-CoV-2 infections. Chronic exposure to immunosuppression, incidence of delayed allograft function, and rejection rates were comparable. However, after one year, COV-KTR showed significantly higher median serum creatinine than Pre-COV-KTR. Our data confirm that KT practice could be safely maintained during the COVID-19 pandemic, with excellent patient- and allograft-related outcomes. Strict infection control strategies, aggressive follow-up monitoring, and preservation of dedicated personnel and resources are key factors for the optimisation of the results in case of future pandemics.

## 1. Introduction

The first wave of coronavirus disease 2019 (COVID-19) pandemic had a catastrophic impact on almost all the healthcare systems worldwide. This was particularly true for those countries where the burden of contagion had exceeded the national reception and treatment capacity of sanitary facilities [1]. Suddenly, we witnessed a generalised reduction in non-COVID-related medical activities, affecting both elective and emergency procedures [2]. Major consequences were observed especially in surgical practice [3]. Key factors were the lack of beds in internal medicine wards and intensive care units (ICU) [4], the scarcity of mechanical ventilators [5], the postponement of diagnostic tests [6,7], the reduced number of operating-room specialists [8], and the poor optimisation of theatre slots [9,10].

The transplant community faced an unprecedented situation posing specific ethical dilemmas and logistical difficulties. The primary discussion mostly focused on the evaluation of the risk–benefit ratio of the transplant activity in the context of a global pandemic caused by a potentially lethal pathogen without any available prophylaxis or treatment [11,12], relying only on the diligent use of non-pharmaceutical interventions (NPI) [13]. This debate particularly applied to kidney transplantation (KT), because dialysis can indefinitely delay surgery without immediate consequences for the patient. Many national and international KT centres stopped their programmes, at least temporarily, to avoid dealing with an unmeasurable hazard or because they could not promptly reorganise and adapt to the new global scenario [8,14,15]. Frequently, the continuation or resumption of the transplant activity was associated with a significant reduction in the number and complexity of the procedures performed [16].

Our unit (in Milan, Italy) did not attempt to interrupt the KT service, avoiding restrictions on the type or complexity of the transplants performed, despite countless difficulties. The aim of the present study was to critically review the results of the KT activity during the COVID-19 pandemic. More specifically, we analysed data from kidney transplant recipients (KTR) and patients on the transplant waiting list (TWL).

## 2. Results

### 2.1. Transplant Waiting-List

#### 2.1.1. Demographic and Clinical Characteristics of Patients on the Transplant Waiting List

The total population of patients on the TWL at our centre between 1 January 2018 and 31 December 2021 included 560 subjects. Among these patients, 203 were enlisted before January 2018, 224 between January 2018 and January 2020, and 133 between January 2020 and December 2021 (COVID era).

Comparing demographic and clinical characteristics of patients in Pre-COV-TWL and COV-TWL, we observed that the two groups were overall similar. Both populations showed a preponderance of male (Pre-COV-TWL: 58.5% vs. COV-TWL: 61.4%; *p* = 0.477) and Caucasian (Pre-COV-TWL: 85.7% vs. COV-TWL: 86.1%; *p* = 0.920) subjects. The median age was also equivalent (Pre-COV-TWL: 52, IQR 44–61 vs. COV-TWL: 53, IQR 45–62; *p* = 0.171). The prevalence of hypertension, diabetes mellitus (DM), chronic obstructive pulmonary disease (COPD), coronary artery disease (CAD), or obesity, as well as the distribution of primary renal diseases, were not significantly different.

Haemodialysis was the most represented form of renal replacement therapy (RRT) and the proportion of dialysis vintage was comparable. As expected, considering the arbitrary censoring of the observation period, the median duration of the follow-up was significantly longer in the COVID group (Pre-COV-TWL: 12 months, IQR 6–24 vs. COV-TWL: 15, IQR 6–24; *p* < 0.001).

Baseline characteristics of patients on the TWL are detailed in Table 1.

#### 2.1.2. Outcomes of Patients on the Transplant Waiting List

The main outcome measures for patients on the TWL were all-cause mortality, definitive exclusion from the TWL (due to death, transplantation, or worsening clinical conditions), and SARS-CoV-2 infection. The follow-up was arbitrarily interrupted in case of death, transplantation, or dropout.

Among COV-TWL patients, we recorded 48 episodes of SARS-CoV-2 infection (48/396, 12.1%), with a lethality rate of 14.6% (7/48). The COVID-19-specific mortality rate in this group was 1.77% (7/396).

Overall, a total of 29 deaths were observed: 10 in Pre-COV-TWL (10/427, 2.3%) and 19 in COV-TWL (19/396, 4.8%). Specifically, the causes of death in the Pre-COV-TWL group were: cardiovascular event (n = 7), septic shock (n = 1), renal cell carcinoma (n = 1), and clinical complications of aortic valve replacement surgery (n = 1). During the pandemic, the causes of death were: SARS-CoV-2 infection (n = 7), cardiovascular event (n = 5), septic shock (n = 4), liver failure (n = 1), complications of limb amputation surgery (n = 1), and hyperkalaemia (n = 1). Although the difference was not statistically significant, there was a trend toward increase mortality during the COVID-19 pandemic (4.8% vs. 2.3%; *p* = 0.061). Remarkably, SARS-CoV-2 infection was responsible for seven out of 19 deaths (36.8%).

Comparing the time spent on the TWL, we observed that the dropout rates were very similar in the two groups (Figure 1).

### 2.2. Kidney Transplants

#### 2.2.1. Donors’ Characteristics

Overall, donor populations included 245 subjects (Pre-COV: 122 vs. COV: 123). Since 23 February 2020, when the first official measures for containment and management of the epidemiological emergency from COVID-19 were issued, all donors were systematically screened for SARS-CoV-2 infection. In cases of positivity, the kidneys were considered as unsuitable for transplantation due to the unknown risk of transmission of the disease through the allograft. All potential donors were assessed by Real-Time quantitative PCR (RT-qPCR) on nasopharyngeal swab and chest X-ray. Additional methods such as high-definition chest CT scan (n = 70) and RT-qPCR on bronchoalveolar lavage (n = 90) or lung aspiration (n = 64) were used in selected cases.

Analysis A1 did not show substantial differences between the main demographic and clinical characteristics of the donors recorded before or during the pandemic. Both groups exhibited a preponderance of male (Pre-COV: 55.7% vs. COV: 58.5%; *p* = 0.699) and Caucasian subjects. The median age was also comparable (Pre-COV: 54 years, IQR 44–61 vs. COV: 55 years, IQR 48–62; *p* = 0.419). During the pandemic, we noticed a significant increase in non-Caucasian donors (Pre-COV: 0/122, 0.0%; COV: 9/123, 7.3%; *p* = 0.003).

The proportions of kidneys retrieved from living or deceased donors remained stable over time, including donations after circulatory death (DCD) and expanded criteria donors (ECD).

Cold ischemia time (CIT) was not affected by the ongoing pandemic (Pre-COV: 12 h 40 min, IQR 10.5–15.7 h vs. COV: 13 h 10 min, IQR 10.3–16.8; *p* = 0.456).

As shown in Table 2, the two groups of donors were further assessed using the Kidney Donor Profile Index (KDPI) [17] and the Kidney Donor Risk Index (KDRI) [18]. Available pre-implantation allograft biopsies (Pre-COV: n = 33 vs. COV: n = 35) were also reviewed using the Karpinski score [19]. The analyses did not show any significant differences between the two groups.

Analysis A2, fully reported in Appendix A, did not offer additional meaningful pieces of information.

#### 2.2.2. Recipients’ Characteristics

Overall, KT recipient populations included 245 patients (Pre-COV-KTR: 122 vs. COV-KTR: 123). During the COVID-19 pandemic, all recipients were pre-operatively screened for SARS-CoV-2 infection by RT-qPCR on nasopharyngeal swab and chest X-ray. In cases of positivity, the patient was temporarily removed from the TWL until the infection was clinically resolved and any signs of the disease excluded by RT-qPCR, chest X-ray, high-resolution chest CT scan, and respiratory function tests.

Analysis A1 showed that the main demographic and clinical characteristics of the recipients transplanted before or during the pandemic were very similar. In both cohorts, most patients were male (Pre-COV-KTR: 58.2% vs. COV-KTR: 55.3%; *p* = 0.699) and Caucasian (Pre-COV-KTR: 91.0% vs. COV-KTR: 88.6%; *p* = 0.699). The median age at transplant was 52 years (IQR 45–60) in the Pre-COVID group and 50 years (IQR 39–58) in the COVID group (*p* = 0.115). Primary kidney diseases and major comorbidities were equally distributed. Previous exposure to CMV, EBV, HSV, VZV, HBV, and HCV was also similar.

We could not find substantial differences in the immunological risk profile of the two groups of recipients. Indeed, the proportions of patients with a history of failed transplant, preformed donor-specific antibody (DSA), HLA mismatch > 4, or panel reactive antibody test (PRA) > 50% were equivalent. Pre-COV-KTR and COV-KTR were further compared using the Estimated Post Transplant Survival (EPTS) score (33, IQR 18–58 vs. 23, IQR 12–46; *p* = 0.009) and the Italian Recipient Case Mix Index (3, IQR 2–4 vs. 3, IQR 2–4; *p* = 0.114) [20,21,22].

As detailed in Table 3, during the COVID-19 period, there was a significant reduction in the use of polyclonal anti-thymocyte globulins ATG (Pre-COV-KTR: 61.5% vs. COV-KTR: 47.2%; *p* = 0.029), with an increased administration of monoclonal anti-C5 antibodies (Pre-COV-KTR: 5.7% vs. COV-KTR: 16.3%; *p* = 0.013). On the contrary, maintenance immunosuppression remained consistent across the two periods.

Analysis A2, fully reported in Appendix A, did not offer additional meaningful pieces of information.

#### 2.2.3. Kidney Transplant Outcomes

Comparing the outcomes of patients transplanted before or during the pandemic, we observed equivalent one-year recipient (Figure 2) and death-censored allograft survival rates (Figure 3).

According to Analysis A1, the causes of death among Pre-COV-KTR were: SARS-CoV-2 infection (n = 3), septic shock (n = 2), and central nervous system malignancy (n = 1). In the COV-KTR group, the causes of death were: ischaemic stroke (n = 1) and myocardial infarction (n = 1).

Before the pandemic, the reasons for transplant loss were: post-operative haemorrhage (n = 2), intra-operative iliac artery dissection (n = 1), polyomavirus-associated nephropathy (n = 1), allograft pyelonephritis (n = 1), and surgical complications of a subsequent liver transplant (n = 1). During the pandemic, the causes of allograft failure were: post-operative haemorrhage (n = 1), acute rejection (n = 1), large B-cell lymphoma (n = 1), urothelial carcinoma of the renal pelvis (n = 1), and massive deep vein thrombosis extending to the allograft (n = 1). Overall, we recorded two episodes of primary non-function (PNF) in each group.

During the COVID-19 era, there was a slight increase in the DGF rate (Pre-COV: 22.1% vs. COV: 27.6%; *p* = 0.376), but one-year cumulative rejection rates remained very similar (Figure 4).

The Comprehensive Complication Index (CCI) was 23 (9–42) in the Pre-COVID group and 21 (0–42) in the COVID one (*p* = 0.236). The duration of hospitalisation was similar. The proportion of patients requiring ICU admission was almost equivalent, but the median ICU length of stay was significantly shorter during the pandemic (Pre-COV-KTR: 1, IQR 1-4 vs. COV-KTR: 1, IQR 1-1; *p* = 0.043). Considering the composite endpoint including DGF, acute rejection, and severe surgical complications, there were 81 events in the Pre-COVID group and 87 in the COVID one.

At every time point, we observed that the median SCr was significantly higher in patients transplanted during the pandemic than controls. In particular, one-year SCr was 1.30 (IQR 1.06–1.57) mg/dL in Pre-COV-KTR and 1.46 (IQR 1.18–2.01) mg/dL in COV-KTR (*p* = 0.008). Transplant-related outcomes according to Analysis A1 are summarised in Table 4.

Analysis A2 did not offer additional meaningful pieces of information (Appendix A).

Overall, 41 cases of SARS-CoV-2 infection were recorded: 19 in the group of patients transplanted before the pandemic and 22 among recipients in the COVID group. The SARS-CoV-2-related mortality rate in KT recipients was 7.3% (3/41). Comparing demographic and clinical characteristics of infected and non-infected recipients, we observed an association between male sex and post-transplant SARS-CoV-2 infection (*p* = 0.009). Available data on SARS-CoV-2 infection are summarised in Table 5 and Table 6, and Appendix A.

To further assess the risk–benefit ratio of transplantation vs. dialysis during the pandemic, we compared one-year survival rates between patients who were TWL and KT recipients. As shown in Figure 5, the results were equivalent.

## 3. Discussion

The first wave of the COVID-19 pandemic represented a strenuous challenge for the international scientific community [23,24], with all the most heavily affected countries witnessing a dramatic collapse of their healthcare systems [1,25]. The inability to meet the increased demand for medical assistance rapidly determined a massive reduction in the volume and quality of the services provided [26,27,28]. As an ultra-specialised multidisciplinary activity involving complex patients and mostly operating in a non-elective setting, solid organs transplantation particularly suffered the global health crisis [14,29]. For many transplant units, especially those located at the epicentre of the pandemic, interrupting the service, or selectively reducing the number of procedures performed, appeared the safest choice [14]. These cautious restraining measures were followed by a progressive resumption of the transplant programmes providing non-replaceable, life-saving organs such as heart, lung, or liver [12,30,31]. On the contrary, the reactions observed among the KT community were extremely heterogeneous [32]. Indeed, the risks associated with hospitalisation, surgery, and immunosuppression in the context of a global pandemic had to be weighed against the unique opportunity to remain on dialysis safely and indefinitely [33]. Although some KT transplant centres preferred to stop their services [34,35], most units decided to adopt specific limitations, aiming to obtain a more favourable balance between risks and benefits. The main options were to restrict transplant procedures to recipients with a low surgical, anaesthesiologic, and immunological risk profile or, alternatively, to life-threatened patients without dialysis access options [36]. Our team maintained the KT programme throughout the entire course of the pandemic, without formal restrictions, striving to offer a high-quality service to all the patients registered on the TWL. The most critical issues addressed while pursuing such goals were: (1) the development of an effective screening programme for SARS-CoV-2 infection for both donors and recipients; (2) the definition of a systematic surveillance strategy for SARS-CoV-2 infection for inpatients, outpatients, and healthcare workers; (3) the policy regulating donors and recipients selection; (4) the choice of the most suitable immunosuppressive induction and maintenance scheme; (5) the scarcity of theatre slots, ICU beds, dialysis machines, and dedicated personnel, resulting in procedural delays and sub-optimal peri-operative care; and (6) the organisation of post-transplant follow-up clinics. Considering the deontological, ethical, and medico-legal implications as much as the need for data possibly guiding future operational strategies and patient counselling, we decided to critically review our performance during the pandemic.

Comparing the demographic and clinical characteristics of the patients registered on the TWL before or during the COVID-10 outbreak, we could not find any substantial difference between the two populations. Moreover, our cohort of end-stage renal disease (ESRD) patients was overall similar to those described in other reports [37,38]. Such observation confirms that, in line with our principles, we did not select transplant candidates aiming to exclude the most complex and frail ones. Furthermore, the homogeneity of the populations analysed grants a more confident interpretation of the results.

Despite the systematic implementation of rigorous safety measures including periodic screening with RT-qPCR on nasopharyngeal swabs of both patients and healthcare workers, access-controlled areas for dialysis-related procedures, strict hygiene standards, and appropriate use of NPI, 12% of our TWL population tested positive for SARS-CoV-2 [39,40]. Available data on the prevalence of SARS-CoV-2 infection among dialysis patients remain conflicting as they were greatly influenced by specific pandemic trends in different geographic regions, local distribution of RRT modalities (peritoneal dialysis vs. haemodialysis), and access to home-dialysis services. Nevertheless, the numbers herein reported are comparable to those observed in studies performed in similar settings (10–15%), basically characterised by very high COVID-19 prevalence in the general population, overcrowded healthcare facilities, scarce development of home-based dialysis programmes, and a striking predominance of in-centre haemodialysis over peritoneal dialysis [41].

In this series, the mortality rate of wait-listed patients with SARS-CoV-2 infection was 15%. The current literature reports mixed results, with overall mortality rates ranging from 10% to 30%. Such a discrepancy between studies makes direct comparisons difficult to interpret [42]. However, the increase in overall mortality observed among our ESRD patients during the pandemic (namely, from 2% to 4%) should raise concern regarding the potential impact of SARS-CoV-2 infection in this particular subset of patients. As a matter of fact, the difference in mortality could be entirely attributed to COVID-19-related deaths, representing half of the total events recorded.

As reported for the TWL population, baseline characteristics of recipients transplanted before or during the COVID-19 pandemic were overall similar and did not show differences that might suggest selection bias or substantial changes in the criteria adopted for transplant eligibility. In our opinion, considering the relatively small sample size, the discrepancy in the prevalence of diabetes mellitus between the two groups could be purely due to chance.

Main donor characteristics (including donor type) also remained consistent over time. The increased number of non-Caucasian and non-Afro-Caribbean donors recorded during the pandemic is difficult to explain and may reflect a relative increase in overall mortality within specific minorities [43].

Remarkably, comparing transplant and recipient immunological risk profiles, no relevant differences could be noticed between the COVID population and the historical control group. Indeed, throughout the pandemic, we continued transplanting marginal kidneys and highly sensitised patients.

Assessing our immunosuppressive strategies, we observed that maintenance regimens were not significantly affected by the pandemic. At every time point of the study, exposure to calcineurin inhibitors (CNI), antiproliferative agents, and steroid daily doses were similar in Pre-COVID and COVID groups. However, during the pandemic, there was a significant decrease in the use of T-cell-depleting globulin. The preferred administration of basiliximab over rATG, especially in the early phase of the pandemic, was driven by the concern that the reduction in circulating lymphocytes could increase the susceptibility to SARS-CoV-2 infection or perhaps infection-related morbidity and mortality. Since the declaration of the pandemic, the choice of the optimal induction and maintenance schemes has represented a major issue for the transplant community [44]. In this regard, it is worth considering that through the first year of the pandemic, there were no approved SARS-CoV-2-specific treatments. Moreover, the vaccination campaign was initiated at the end of 2020. Due to the lack of formal clinical guidelines and the scarcity of available data, the management of immunosuppression remained at the discretion of the single transplant units. Most centres opted for an anti-IL-2 receptor antagonist induction associated with a triple-agent maintenance scheme based on tacrolimus, MMF/MPA, and steroids [45]. The common trend was to progressively reduce the net state of immunosuppression tapering CNIs or antiproliferative compounds [46]. In some cases, MMF or MPA were empirically replaced by mTOR inhibitors, relying on their recognised antiviral properties against EBV, CMV, and BKV [47,48]. To date, the real impact of the strategies on the incidence and severity of post-transplant SARS-CoV-2 infections, as well as their effects on rejection rates and allograft survival, remains undetermined as no randomised or parallel groups studies have been published [49].

One-year recipient and allograft survivals during the pandemic, the primary outcomes of our analysis, were excellent and in line with current international standards. Importantly, both parameters were identical to those recorded before the pandemic. Moreover, no recipients died from SARS-CoV-2 in the COVID-19 period.

Although not statistically significant, we observed an increase in the incidence of DGF (22% vs. 28%) during the COVID period. On the contrary, acute rejection rate remained consistent over time. The underlying causes of this relative rise in DGF remain to be defined. Certainly, the rise cannot be ascribed to differences in donors’ characteristics, organs quality, or CIT, as these parameters did not substantially change during the COVID-19 era. Unfortunately, there are no data on warm ischaemia times related to organ procurement or transplantation, thus their actual role is undeterminable. During the pandemic, we noticed a lower threshold for early post-transplant dialysis as our nephrologists were seriously concerned regarding the prompt availability of dialysis beds in case of urgent need.

One-year serum creatinine, a surrogate marker of long-term allograft function and survival, was slightly higher among patients transplanted during the COVID period than controls (1.30 vs. 1.46 mg/dL). This finding is difficult to explain as donor-, transplant-, and recipient-related characteristics, as well as the KDPI, KDRI, pre-implantation Karpinski score, EPTS, and Italian Recipient Case Mix Index of Pre-COVID and COVID KT patients, were overall similar. Other variables, more difficult to measure, may have contributed, including a reduction in outpatient follow-up visits, a less timely diagnosis of adverse events, and the resulting delay in care. This was particularly true during the very early stages of the pandemic, when most patients hesitated to attend hospital care due to the perceived risk of contagion, and when the number of active members of the nephrology team dedicated to outpatient clinics was cutdown. As a matter of fact, many nephrologists previously involved in outpatient post-KT follow-up activities were employed in extraordinary tasks in accident and emergency departments or COVID-19 wards. Moreover, the occurrence of SARS-CoV-2 infection may have played a role as COVID-19 has been associated with acute kidney injury and irreversible loss of renal function in the general population and transplanted subjects [50,51]. The short- and long-term effects of immunosuppression minimisation in the case of subclinical or overt disease should also be considered as much as the discretional use of basiliximab over rATG observed at the beginning of the pandemic. In fact, both factors might have determined an increase in subclinical rejection episodes, with associated chronic allograft damage. Finally, we believe that the re-organisation of the non-elective surgical activity as much as the very early post-operative care during the pandemic peak could have caused an increase in post-transplant surgical and medical complications. Undoubtedly, due to the scarcity of personnel available, we often had to operate out of hours, in non-dedicated theatres, and with anaesthesiologists and scrub and ward nurses lacking in specific transplant expertise.

The incidence of SARS-CoV-2 infection among patients transplanted during the pandemic at our centre was 17%, with an overall COVID-19-related mortality of 1% and a lethality rate of 8%. Once again, comparison with other reports is extremely complex due to the generalised paucity and heterogeneity of studies [52,53,54]. However, our findings confirm that KT recipients with COVID-19 have a relatively low mortality and a better prognosis than ESRD patients remining on dialysis. Moreover, while supporting systematic implementation of SARS-CoV-2 infection control and management strategies, these data fundamentally reassure both transplant clinicians and transplant candidates.

The present study has, at least, two major limitations including the relatively small sample size and the short duration of the follow-up. Nevertheless, the experience herein reported represents a rare contribution to the existing literature and can validly support the maintenance of KT programmes.

In conclusion, our data demonstrate that the COVID-19 pandemic was not associated with inferior short-term KT recipient or allograft survivals compared to the pre-pandemic era. We observed that, during the SARS-CoV-2 outbreak, the survival of KT recipients remained substantially unchanged. On the contrary, patients on the TWL experienced an increase in mortality, mostly due to episodes of lethal COVID-19. Overall, these findings strongly support the maintenance of KT programmes, also in the event of a generalised health crisis. The observation that one-year allograft function was slightly better in the group of patients transplanted before the pandemic warrants further investigation to rule out possible long-term effects on transplant survival. Moreover, it prompts exploration of less measurable pandemic-related factors possibly affecting transplant activity and outcomes. The importance of high-quality specialised surgical and medical care during the peri-operative and post-transplant follow-up cannot be emphasised enough.

## 4. Materials and Methods

In this single-centre retrospective observational study, we enrolled 560 adult patients with ESRD registered on the kidney TWL of the Fondazione IRCCS Ca’ Granda Ospedale Maggiore Policlinico in Milan (Italy), between January 2018 and December 2020. We decided to exclude paediatric subjects (n = 83) because they were followed up in another facility by a dedicated nephrology team. Among included patients, 245 received a KT during the study period. For analysis purposes, the entire population was classified into different groups (patients on the transplant waiting list, TWL vs. kidney transplant recipients, KTR) and subgroups (patients enlisted or transplanted before the COVID-19 pandemic, Pre-COV vs. those enlisted or transplanted during the pandemic, COV). Considering the peculiar SARS-CoV-2 epidemiology in the Lombardy region, as a start date of the COVID-19 period, we arbitrarily chose 1 January 2020. Patients were considered as Pre-COV or COV accordingly. The follow-up was intentionally interrupted on 31 December 2021, or in case of specific events such as death, definitive suspension from the TWL, transplantation, or allograft loss. Data were collected using national, regional, and local sources, as well as institutional medical records. The flow diagram of the study is depicted in Figure 6.

For each patient on the TWL, we collected demographic data (age, sex, ethnicity), comorbidities (hypertension, DM, CAD, COPD, obesity), renal history (primary renal disease, previous transplants, type of renal replacement therapy), and infections (CMV, EBV, HSV, VZV, HBV, or HCV), with detailed information on SARS-CoV-2.

Donor-related data included demographic characteristics (age, sex, ethnicity), type of donation (DBD, DCD, ECD, or living donation), comorbidities (arterial hypertension, DM, cerebrovascular disease), renal function (as assessed by serum creatinine), CIT, KDPI, KDRI, time-0 allograft biopsy (Karpinski score), and screening for SARS-CoV-2 infection at the time of organ procurement. For deceased donors, the length of stay in ICU before the declaration of death was also considered.

The recipients were characterised using information retrieved during pre-transplant assessment, transplant-related admission, and outpatient post-transplant follow-up. The EPTS score and the Italian Recipient Case Mix Index were also calculated.

KT-specific data included donor-recipient ABO and HLA compatibility, last and maximum PRA test, circulating DSA at the time of transplant, immunosuppression (schemes and exposure), length of hospitalisation, ICU admission, PNF, DGF, allograft rejection, post-operative complications with associated treatments, episodes of SARS-CoV-2 infection, patient and allograft survivals, and renal function (at discharge, one, three, six, nine, and 12 months after transplant).

As induction therapy, low-immunological risk recipients (first transplant, last PRA < 50%, no preformed DSA, and living or standard-criteria DBD donor) were administered intravenous (IV) basiliximab (Simulect^®^, Novartis, Basel, Switzerland) 20 mg at the time of transplant and on post-operative day 4. High-immunological risk patients received IV rATG (Thymoglobulin^®^, Genzyme, Cambridge, MA, USA) 1 mg/kg/total dose from day 0 to day 4. All subjects were given IV methylprednisolone 500 mg on day 0 and 125 mg on day 1 and day 2. Transplant candidates with circulating DSA > 3000 mean fluorescence intensity (MFI) underwent pre- and post-operative plasma exchange with fresh frozen plasma/albumin and received IV human polyclonal immunoglobulin type G (IgG) 2 g/kg total dose. A few heavily sensitised patients were also treated with anti-CD20 monoclonal antibodies. Recipients with atypical haemolytic uremic syndrome additionally received IV eculizumab (Soliris^®^, Alexion, Boston, MA, USA) 900-to-1200 mg immediately before surgery.

Maintenance immunosuppression consisted of a triple-agent scheme including oral tacrolimus, MMF/MPA, and steroids. Tacrolimus (Adoport^®^, Novartis, Basel, Switzerland; Envarsus^®^, Veloxis Pharmaceuticals, Hørsholm, Denmark; or Advagraf^®^, Astellas Pharma, Chuo City, Tokyo, Japan) was administered and adjusted to achieve a trough level of 8–12 ng/mL during the first month and 6-8 ng/mL thereafter. Patients were also given MMF (Myfenax^®^, Teva, Petach Tikva, Israel) 2000 mg/day or MPA (Myfortic^®^, Novartis, Basel, Switzerland) 1440 mg/day. The dose was reduced by 50% after six months of follow-up. Prednisone was administered 20 mg/day from post-operative day 3 and progressively tapered to 5 mg/day by post-transplant day 30.

As a prophylaxis for Pneumocystis jirovecii pneumonia, we used oral trimethoprim-sulfamethoxazole 80/400 mg every other day for three months. Recipients at increased risk of CMV disease received oral valganciclovir for three-to-six months, with the dose adjusted according to renal function.

The main goal of the present study was to assess the impact of the COVID-19 pandemic on our KT practice, particularly focusing on safety. Accordingly, the primary outcomes were recipient mortality (overall and cause-specific), allograft loss, and SARS-CoV-2-related adverse events. After comparing the results of KT performed before or during the pandemic, we aimed to compare safety parameters (namely, mortality and SARS-CoV-2-related morbidity) between patients remaining on the TWL or transplanted during the pandemic. As secondary safety measures, the following variables were considered: PNF, DGF, rejection, post-transplant complications, hospital length of stay, and re-hospitalisation. Patients were diagnosed PNF in the case of allograft function being unable to prevent continuous RRT where other possible complications were ruled out. DGF was defined as the need for dialysis during the first post-transplant week. We considered allograft function (serum creatinine at discharge, one, three, six, nine, and 12 months of follow-up) and exposure to immunosuppression (tacrolimus tough level, daily MMF/MPA dose, and daily steroid dose) as efficacy parameters. Finally, we compared the mortality (absolute and cause-specific) and transplant rate (expressed as the proportion of patients receiving a KT in a prespecified time interval) of subjects registered on the TWL before or during the COVID-19 pandemic.

Dichotomous variables were described using absolute numbers and proportions (%). Numerical variables were represented as medians and interquartile ranges (IQR). Results were compared using Fisher’s exact test or the Mann–Whitney U-test, as appropriate. Time-dependent variables were analysed by the Kaplan–Meier method. Survival curves were compared with the log-rank test. To analyse the variables of interest as comprehensively as possible, we developed two statistical models. The first analysis (Analysis A1) primarily focused on patients on the TWL. Data from subjects enlisted before the pandemic (from January 2018 to December 2019) were compared with those awaiting a KT during the pandemic (from January 2020 to December 2021). Two analyses were performed on KT recipients. In Analysis A1, data from patients who underwent transplantation in the Pre-COVID-19 period (from January 2018 to December 2019) were compared with those transplanted during the pandemic (from January 2020 to December 2021). In Analysis A2, we compared data from patients undergoing KT before the pandemic (from January 2018 to December 2019, with arbitrary censoring of follow-up on 31 December 2019) with those from patients transplanted during the pandemic (from January 2020 to December 2021) or who had undergone KT before the pandemic, but still had a functioning allograft during the COVID-19 period. This second model was basically introduced to account for the effects of the COVID-19 pandemic on the entirety of our KT population. Finally, we compared data from patients transplanted during the COVID-19 period (from January 2020 to December 2021) with those of patients remaining on the TWL in the same period. Significance of the statistical tests was retained when *p* < 0.05. Analyses were performed using SPSS (version 25.0; IBM Corp., Armonk, NY, USA).

Treatments and procedures herein reported were in accordance with the ethical standards of the institutional committee at which it was conducted (Fondazione IRCCS Ca’ Granda Ospedale Maggiore Policlinico Ethical Committee), as well as with the 1964 Helsinki declaration and its later amendments, or comparable ethical standards. All participants included in the study consented to enlistment in the TWL, KT, treatments, and follow-up investigations. A specific consent for data collection and analysis was obtained from all the subjects referred to our hospital during the COVID-19 pandemic. As a retrospective observational (non-interventional) study involving KT and using an anonymised dataset, the present work refers to the institutional Protocol ID 4759-1837/19.

## Figures and Tables

**Figure 1 pathogens-11-01144-f001:**
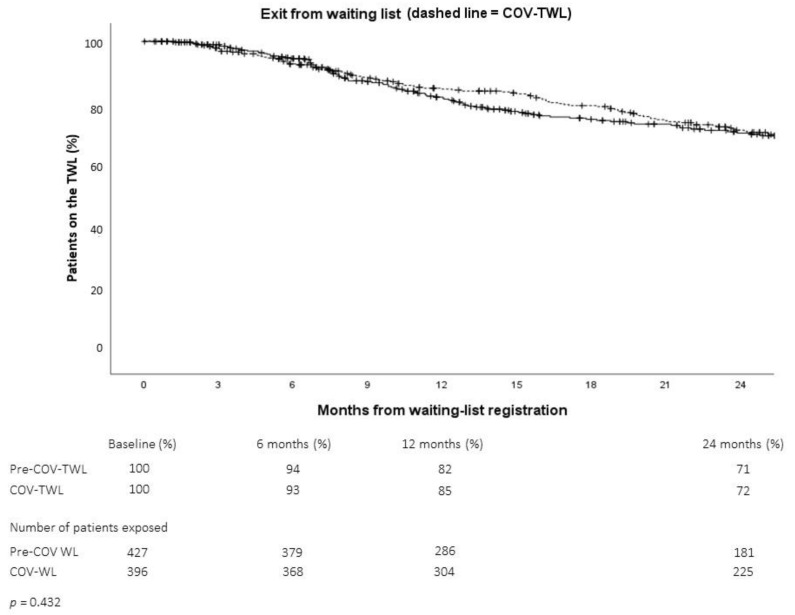
Dropout rates of patients on the kidney transplant waiting list (TWL) before (Pre-COV, solid line) or during (COV, dashed line) the COVID-19 pandemic.

**Figure 2 pathogens-11-01144-f002:**
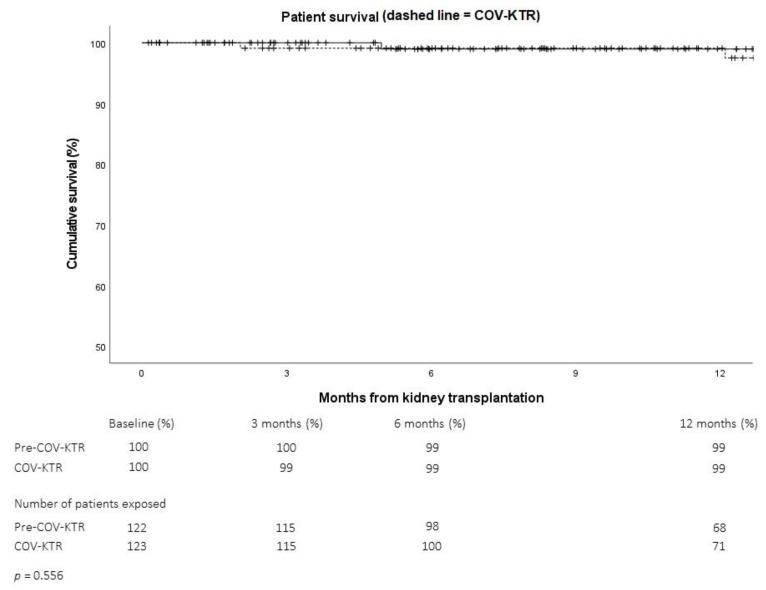
One-year kidney transplant recipient survival rate before (Pre-COV-KTR, solid line) or during (COV-KTR, dashed line) the COVID-19 pandemic (Analysis A1).

**Figure 3 pathogens-11-01144-f003:**
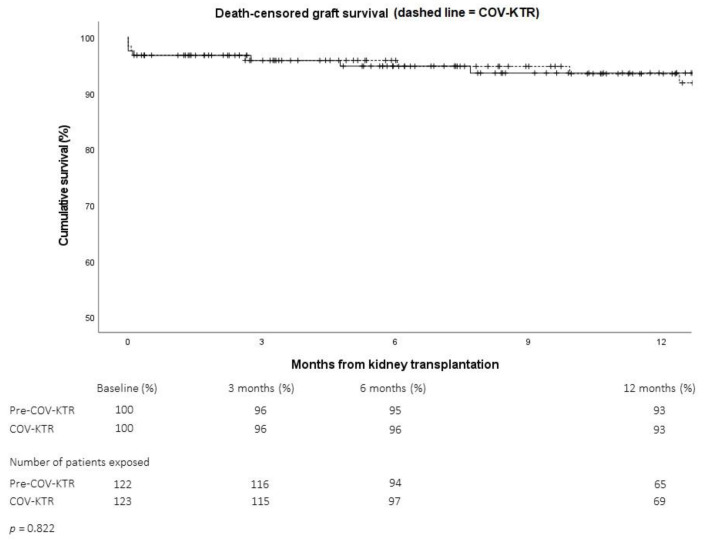
One-year death-censored kidney allograft survival rate before (Pre-COV-KTR, solid line) or during (COV-KTR, dashed line) the COVID-19 pandemic (Analysis A1).

**Figure 4 pathogens-11-01144-f004:**
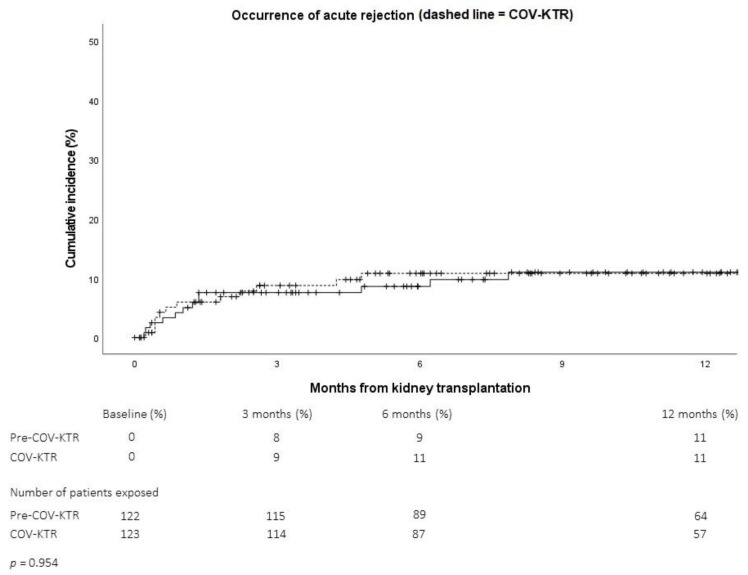
One-year cumulative acute rejection rate before (Pre-COV-KTR, solid line) or during (COV-KTR, dashed line) the COVID-19 pandemic (Analysis A1).

**Figure 5 pathogens-11-01144-f005:**
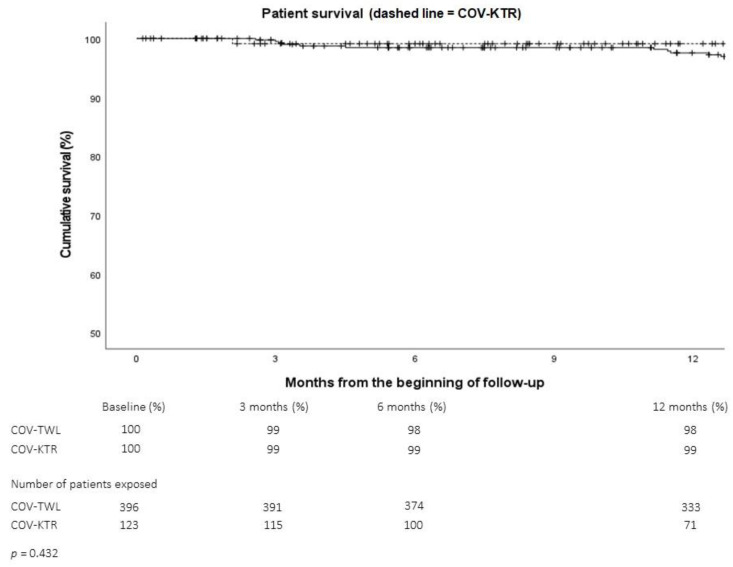
Survival rates of patients remaining on the transplant waiting list (COV-TWL, solid line) or transplanted (COV-KTR, dashed line) during the COVID-19 pandemic.

**Figure 6 pathogens-11-01144-f006:**
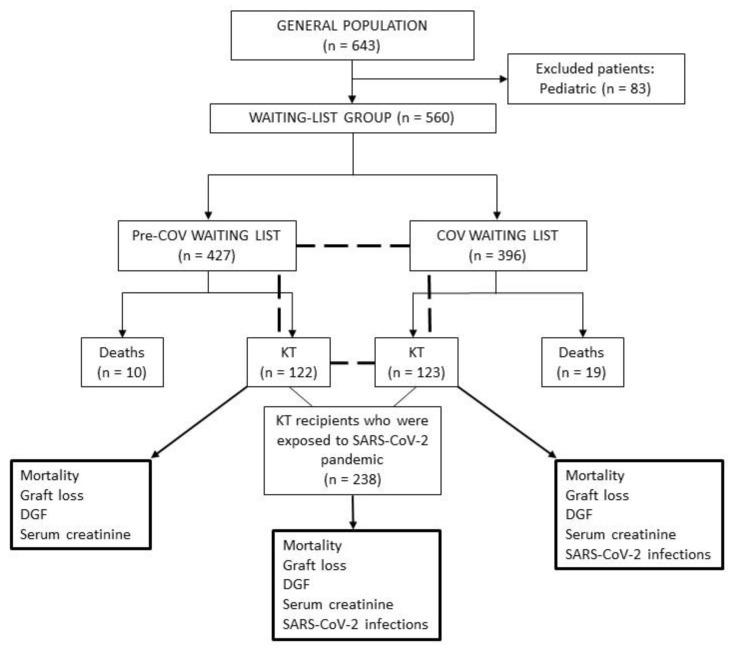
Flow diagram of the study.

**Table 1 pathogens-11-01144-t001:** Demographic and clinical characteristics of patients on the kidney transplant waiting list (TWL) before (Pre-COV) or during (COV) the COVID-19 pandemic.

Variables	Whole Population (N = 823)	Pre-COV-TWL (N = 427)	COV-TWL (N = 396)	*p*
Sex (males)	494 (60.0)	251 (58.5)	243 (61.4)	0.477
Age (years)	53 (44–61)	52 (44–61)	53 (45–62)	0.171
Ethnicity:				
Caucasian	707 (85.9)	366 (85.7)	341 (86.1)	0.920
Afro-Caribbean	40 (4.9)	20 (4.7)	20 (5.1)	0.872
Other	76 (9.2)	41 (9.6)	35 (8.8)	0.720
Renal replacement therapy	741 (90.0)	388 (90.9)	353 (89.1)	0.418
Haemodialysis	590/741 (79.6)	310/388 (79.9)	280/353 (79.3)	0.856
Dialysis vintage (months)	37 (17–59)	34 (16–57)	40 (19–62)	0.074
Previous kidney transplant	200 (24.3)	111 (26.0)	89 (22.5)	0.256
Primary kidney disease:				
Primary or secondary glomerulonephritis	417 (50.7)	210 (49.2)	207 (52.3)	0.403
Diabetic nephropathy	41 (5.0)	19 (4.4)	22 (5.6)	0.523
Polycystic kidney disease	112 (13.6)	61 (14.3)	51 (12.9)	0.611
Hypertensive nephrosclerosis	81 (9.8)	35 (8.2)	46 (11.6)	0.103
Tubulo-interstitial disease	37 (4.5)	19 (4.4)	18 (4.5)	1.000
Genetic kidney disease	35 (4.3)	21 (4.9)	14 (3.5)	0.389
Uropathy	84 (10.2)	45 (10.5)	39 (9.8)	0.818
Thrombotic microangiopathy	40 (4.9)	21 (4.9)	19 (4.8)	1.000
Ischaemia	7 (0.9)	5 (1.2)	2 (0.5)	0.452
Unknown	17 (2.1)	14 (3.3)	3 (0.8)	0.013
Pre-existing conditions:				
Arterial hypertension	713 (88.5)	363 (87.9)	350 (89.1)	0.660
Diabetes mellitus	103 (12.8)	53 (12.8)	50 (12.7)	1.000
Chronic obstructive pulmonary disease	144 (17.9)	74 (17.9)	70 (17.8)	1.000
Coronary artery disease	134 (16.6)	67 (16.2)	67 (17.1)	0.777
Obesity (BMI ≥ 30 kg/m^2^)	90 (11.2)	45 (10.9)	45 (11.5)	0.824
CMV IgG positivity	701 (87.1)	366 (88.6)	335 (85.5)	0.207
EBV IgG positivity	727 (90.3)	375 (90.8)	352 (89.8)	0.636
HSV IgG positivity	655 (81.4)	339 (82.1)	316 (80.6)	0.651
VZV IgG positivity	778 (96.6)	401 (97.1)	377 (96.2)	0.558
HBV viremia	8 (1.0)	4 (1.0)	4 (1.0)	1.000
HCV viremia	14 (1.7)	8 (1.9)	6 (1.5)	0.790
Follow-up (months)	13 (6–24)	12 (6–24)	15 (6–24)	<0.001

Abbreviations: BMI, body mass index; CMV, cytomegalovirus; EBV, Epstein–Barr virus; HBV, hepatitis B virus; HCV, hepatitis C virus; HSV, herpes simplex virus; IgG, immunoglobulin G; VZV, varicella-zoster virus.

**Table 2 pathogens-11-01144-t002:** Demographic and clinical characteristics of kidney transplant donors before (Pre-COV) or during (COV) the COVID-19 pandemic (Analysis A1).

Variables	Whole Population (N = 245)	Pre-COV Donors (N = 122)	COV Donors (N = 123)	*p*
Donor sex (male)	140 (57.1)	68 (55.7)	72 (58.5)	0.699
Donor age (years)	55 (46–61)	54 (44–61)	55 (48–62)	0.419
Type of donor:				
DBD	178 (72.7)	82 (67.2)	96 (78.0)	0.063
DCD	19 (7.8)	13 (10.7)	6 (4.9)	0.100
ECD	98 (40.0)	48 (39.3)	50 (40.7)	0.896
LD	48 (19.6)	27 (22.1)	21 (17.1)	0.338
Donor risk factors:				
Cerebrovascular accident	99 (40.4)	26 (37.7)	53 (43.1)	0.435
Arterial hypertension	79 (32.2)	37 (30.3)	42 (34.1)	0.585
Last SCr > 1.5 mg/dL	32 (13.1)	18 (14.8)	14 (11.4)	0.455
ICU admission	197 (80.4)	95 (77.9)	102 (82.9)	0.338
ICU stay (days)	3 (1–5)	3 (1–5)	3 (2–5)	0.885
Donor ethnicity:				
Caucasian	236 (96.3)	122 (100)	114 (92.7)	0.003
Afro-Caribbean	1 (0.4)	0 (0.0)	1 (0.8)	1.000
Other	8 (3.3)	0 (0.0)	8 (6.5)	0.007
Cold ischemia time (minutes)	780 (623–960)	760 (630–940)	790 (620–1010)	0.456
KDPI	63 (34–85)	61 (31–84)	63 (35–87)	0.471
KDRI	1.12 (0.85–1.45)	1.12 (0.82–1.44)	1.12 (0.86–1.49)	0.558
Karpinski score *	3 (3–4)	4 (3–4)	3 (3–4)	0.416

* The Karpinski score was available for 33 donors in the Pre-COVID group and 35 donors in the COVID group. Abbreviations: DBD, donor after brain death; DCD, donor after circulatory death; ECD, expanded criteria donor; ICU, intensive care unit; KDPI, kidney donor profile index; KDRI, kidney donor risk index; LD, living donor; SCr, serum creatinine.

**Table 3 pathogens-11-01144-t003:** Demographic and clinical characteristics of kidney transplant recipients (KTR) before (Pre-COV-KTR) or during (COV-KTR) the COVID-19 pandemic (Analysis A1).

Variables	Whole Population (N = 245)	Pre-COV-KTR (N = 122)	COV-KTR (N = 123)	*p*
Recipient sex (male)	139 (56.7)	71 (58.2)	68 (55.3)	0.699
Recipient age (years)	52 (43–59)	52 (45–60)	50 (39–58)	0.115
Recipient ethnicity:				
Caucasian	220 (89.8)	111 (91.0)	109 (88.6)	0.674
Afro-Caribbean	6 (2.4)	1 (0.8)	5 (4.1)	0.213
Other	19 (7.8)	10 (8.2)	9 (7.3)	0.816
Renal replacement therapy	230 (96.7)	118 (96.7)	112 (91.1)	0.107
Haemodialysis	189/230 (82.2)	92/118 (78.0)	97/112 (86.6)	0.120
Dialysis vintage (months)	35 (16–56)	34 (16–62)	37 (15–56)	0.728
Primary kidney disease:				
Primary or secondary glomerulonephritis	113 (46.1)	56 (45.9)	57 (46.3)	1.000
Diabetic nephropathy	8 (3.3)	5 (4.1)	3 (2.4)	0.500
Polycystic kidney disease	41 (16.7)	25 (20.5)	16 (13.0)	0.127
Hypertensive nephrosclerosis	17 (6.9)	7 (5.7)	10 (8.1)	0.616
Tubulointerstitial disease	9 (3.7)	3 (2.5)	6 (4.9)	0.500
Genetic or congenital kidney disease	27 (11.0)	13 (10.7)	14 (11.4)	1.000
Uropathy	30 (12.2)	12 (9.8)	18 (14.6)	0.330
Thrombotic microangiopathy	17 (6.9)	7 (5.7)	10 (8.1)	0.616
Other	3 (1.2)	2 (1.6)	1 (0.8)	0.622
Pre-existing comorbidities:				
Arterial hypertension	211 (86.1)	106 (86.9)	105 (85.4)	0.854
Diabetes mellitus	24 (9.8)	19 (15.6)	5 (4.1)	0.002
Chronic obstructive pulmonary disease	34 (13.9)	19 (15.6)	15 (12.2)	0.466
Coronary artery disease	29 (11.8)	11 (9.0)	18 (14.6)	0.235
Obesity (BMI ≥ 30 kg/m^2^)	17 (6.9)	10 (8.2)	7 (5.7)	0.464
CMV IgG positivity	209 (85.3)	109 (89.3)	100 (81.3)	0.104
EBV IgG positivity	212 (86.5)	107 (87.7)	105 (85.4)	0.709
HSV IgG positivity	196 (80.0)	96 (78.7)	100 (81.3)	0.635
VZV IgG positivity	237 (96.7)	118 (96.7)	119 (96.7)	1.000
HBV viremia	5 (2.0)	1 (0.8)	4 (3.3)	0.370
HCV viremia	3 (1.2)	1 (0.8)	2 (1.6)	1.000
Previous kidney transplant	56 (22.9)	33 (27.0)	23 (18.7)	0.130
Last PRA (%)	0 (0–0)	0 (0–0)	0 (0–0)	0.184
N° Baseline DSA	0 (0–0)	0 (0–0)	0 (0–0)	0.095
Baseline DSA	14 (5.7)	10 (8.2)	4 (3.3)	0.107
HLA mismatch	4 (3–5)	4 (3–5)	4 (3–5)	0.076
EPTS score	28 (16–54)	33 (18–58)	23 (12–46)	0.009
Italian Recipient Case Mix Index	3 (2–4)	3 (2–4)	3 (2–4)	0.114
Length of hospitalisation	13 (10–21)	15 (10–20)	13 (10–22)	0.751
ICU admission	58 (23.7)	26 (21.3)	31 (25.2)	0.546
ICU stay (days)	1 (1–2)	1 (1–4)	1 (1–1)	0.043
Induction immunosuppression:				
Anti-IL2R monoclonal antibodies	126 (51.4)	56 (45.9)	70 (56.9)	0.097
rATG	133 (54.3)	75 (61.5)	58 (47.2)	0.029
Methylprednisolone	245 (100)	122 (100)	123 (100)	1.000
Anti-CD20 monoclonal antibodies	15 (6.1)	9 (7.4)	6 (4.9)	0.439
Anti-C5 monoclonal antibodies	27 (11.0)	7 (5.7)	20 (16.3)	0.013
Plasma exchange	23 (9.4)	15 (12.3)	8 (6.5)	0.131
Polyclonal human immunoglobulins	19 (7.8)	13 (10.7)	6 (4.9)	0.100
Maintenance immunosuppression ^#^:				
Tacrolimus	242/242 (100.0)	120/120 (100.0)	122/122 (100.0)	1.000
MMF/MPA	237/242 (97.9)	116/120 (96.7)	121/122 (99.2)	0.211
Prednisone	237/242 (97.9)	117/120 (97.5)	120/122 (98.4)	0.682
Tacrolimus trough levels (ng/mL):				
After 1 month	9.1 (7.5–11.1)	9.1 (7.6–10.9)	9.0 (7.4–13.8)	0.695
After 3 months	8.4 (7.1–10.2)	9.1 (7.1–10.2)	8.2 (7.1–10.0)	0.248
After 6 months	8.4 (6.9–10.0)	8.9 (7.3–10.1)	8.0 (6.7–9.6)	0.090
After 9 months	7.9 (6.7–9.4)	7.7 (6.5–8.6)	8.3 (6.8–9.8)	0.120
After 12 months	7.7 (6.5–9.3)	8.1 (6.8–9.6)	7.5 (6.4–9.1)	0.231
Follow-up (months)	12.2 (5.7–17.8)	12.4 (5.6–17.7)	12.0 (5.9–18.0)	0.889

^#^ Three patients did not receive maintenance immunosuppression: in the first patient, the organ did not show reperfusion after revascularisation and was therefore immediately removed; in the second patient, bleeding from the surgical site in the first post-transplantation hours led to immediate graftectomy; in the third patient, the external iliac artery dissected, which required immediate transplantectomy and bypass surgery. Abbreviations: BMI, body mass index; CMV, cytomegalovirus; DSA, donor-specific antibody; EBV, Epstein–Barr virus; EPTS, estimated post-transplant survival; HBV, hepatitis B virus; HCV, hepatitis C virus; HLA, human leukocyte antigen; HSV, herpes simplex virus; IgG, immunoglobulin G; IL2R, interleukin-2 receptor; MMF/MPA, mycophenolate mofetil/mycophenolic acid; PRA, panel reactive antibody; rATG, rabbit anti-thymocyte globulin; VZV, varicella-zoster virus.

**Table 4 pathogens-11-01144-t004:** One-year kidney transplant-related outcomes before (Pre-COV-KTR) or during (COV-KTR) the COVID-19 pandemic (Analysis A1).

Variables	Whole Population (N = 245)	Pre-COV-KTR (N = 122)	COV-KTR (N = 123)	*p*
PNF	4 (1.2)	2 (1.6)	2 (1.6)	1.000
DGF	61 (24.9)	27 (22.1)	34 (27.6)	0.376
DGF duration (days)	7 (4–12)	7 (5–11)	6 (4–12)	0.641
CCI	23 (0–42)	23 (9–42)	21 (0–42)	0.236
SCr at discharge	1.46 (1.14–1.94)	1.34 (1.10–1.73)	1.60 (1.20–2.08)	0.003
SCr after 1 month	1.5 (1.18–1.88)	1.32 (1.07–1.63)	1.65 (1.28–2.06)	<0.001
SCr after 3 months	1.44 (1.16–1.81)	1.33 (1.11–1.60)	1.61 (1.26–2.06)	<0.001
SCr after 6 months	1.44 (1.17–1.80)	1.32 (1.15–1.60)	1.61 (1.19–2.08)	<0.001
SCr after 9 months	1.41 (1.12–1.78)	1.35 (1.07–1.56)	1.49 (1.18–1.98)	0.003
SCr after 12 months	1.38 (1.12–1.70)	1.30 (1.06–1.57)	1.46 (1.18–2.01)	0.008
N° of hospital re-admissions	0 (0–1)	0 (0–1)	0 (0–1)	0.789
SARS-CoV-2 cases	22 (9.0)	0 (0.0)	22 (17.9)	<0.001

Abbreviations: CCI, comprehensive complication index; DGF, delayed graft function; N°, number; PNF, primary non-function; SCr, serum creatinine.

**Table 5 pathogens-11-01144-t005:** Demographic and clinical characteristics of kidney transplant recipients (KTR) with (COVID) or without (NON-COVID) SARS-CoV-2 infection.

Variables	NON-COVID KTR (N = 204)	COVID KTR (N = 41)	*p*
Recipient sex (male)	108 (52.9)	31 (75.6)	0.009
Recipient age (years)	52 (43–59)	52 (32–61)	0.469
Recipient ethnicity:			
Caucasian	185 (90.7)	35 (85.4)	0.393
Afro-Caribbean	6 (2.9)	0 (0.0)	0.593
Other	13 (6.4)	6 (14.6)	0.102
Renal replacement therapy	193 (94.6)	37 (90.2)	0.288
Haemodialysis	157/193 (81.3)	32/37 (86.5)	0.639
Dialysis vintage (months)	35 (15–54)	44 (21–72)	0.234
Primary kidney disease:			
Primary or secondary glomerulonephritis	90 (44.1)	24 (58.5)	0.122
Diabetic nephropathy	8 (3.9)	0 (0.0)	0.359
Polycystic kidney disease	38 (18.6)	3 (7.3)	0.106
Hypertensive nephrosclerosis	14 (6.9)	3 (7.3)	1.000
Tubulo-interstitial disease	8 (3.9)	1 (2.4)	1.000
Genetic or congenital kidney disease	20 (9.8)	7 (17.1)	0.178
Uropathy	26 (12.7)	4 (9.8)	0.795
Thrombotic microangiopathy	16 (7.8)	1 (2.4)	0.320
Other	3 (1.5)	0 (0.0)	1.000
Pre-existing conditions:			
Arterial hypertension	178 (87.3)	33 (80.5)	0.320
Diabetes mellitus	20 (9.8)	4 (9.8)	1.000
Chronic obstructive pulmonary disease	32 (15.7)	2 (4.9)	0.083
Coronary artery disease	23 (11.3)	6 (14.6)	0.596
Obesity (BMI ≥ 30 kg/m^2^)	15 (7.4)	2 (4.9)	0.745
CMV IgG positivity	173 (84.8)	36 (87.8)	0.810
EBV IgG positivity	180 (88.2)	32 (78.0)	0.128
HSV IgG positivity	166 (81.4)	30 (73.2)	0.283
VZV IgG positivity	198 (97.1)	39 (95.1)	0.624
HBV viremia	3 (1.5)	2 (4.9)	0.196
HCV viremia	2 (1.0)	1 (2.4)	0.424
Previous kidney transplantation	46 (22.5)	10 (24.4)	0.839
Last PRA (%)	0 (0–0)	0 (0–0)	0.411
Baseline DSA	0 (0–0)	0 (0–0)	0.778
HLA mismatch	4 (3–5)	4 (3–5)	0.560
Length of hospitalisation	14 (10–21)	13 (9–23)	0.644
ICU admission ICU stay (days)	46 (22.5) 1 (1–1)	11 (26.8) 1 (1–3)	0.548 0.332
Induction immunosuppression:			
Anti-IL2R monoclonal antibodies	108 (52.9)	20 (48.8)	0.732
rATG	112 (54.9)	21 (51.2)	0.732
Methylprednisolone	203 (99.5)	41 (100)	1.000
Anti-CD20 monoclonal antibodies	10 (4.9)	3 (7.3)	0.461
Anti-C5 monoclonal antibodies	25 (12.3)	2 (4.9)	0.272
Plasma exchange	19 (9.3)	6 (14.6)	0.393
Polyclonal human immunoglobulins	15 (7.4)	4 (9.8)	0.534
Maintenance immunosuppression:			
Tacrolimus	201 (98.5)	41 (100)	1.000
MMF/MPA	194 (95.1)	41 (100)	0.221
Prednisone	196 (96.1)	41 (100)	0.359
Tacrolimus trough levels (ng/mL):			
After 1 month	9.1 (7.7–11.2)	9.2 (7.3–13.4)	0.768
After 3 months	8.3 (7.0–9.8)	9.2 (8.1–11.7)	0.007
After 6 months	8.3 (6.8–10.0)	8.7 (7.8–10.1)	0.359
After 9 months	7.9 (6.7–9.4)	8.1 (6.1–9.5)	0.636
After 12 months	7.8 (6.5–9.2)	7.4 (6.5–9.9)	0.997
Follow-up (months)	20.7 (8.7–36.3)	19.6 (11.3–30.5)	0.920

Abbreviations: BMI, body mass index; CMV, cytomegalovirus; DSA, donor-specific antibody; EBV, Epstein–Barr virus; HBV, hepatitis B virus; HCV, hepatitis C virus; HLA, human leukocyte antigen; HSV, herpes simplex virus; IgG, immunoglobulin G; IL2R, interleukin-2 receptor; MMF/MPA, mycophenolate mofetil/mycophenolic acid; PRA, panel reactive antibody; rATG, rabbit anti-thymocyte globulin; VZV, varicella-zoster virus.

**Table 6 pathogens-11-01144-t006:** Characteristics, management, and outcomes of SARS-CoV-2 infection after kidney transplantation (KT).

Patient	Months from KT	Hospital Admission	Symptoms	Treatment				Recipient Outcome	Allograft Outcome
				CNI ↓	MMF ↓	MP ↑	Other		
1	32	Yes	Pneumonia + RF	Yes	Yes	Yes	Abx, O2	Alive	Stable
2	41	Yes	FLS + RF	No	Yes	Yes	Remdesivir, O2	Alive	Stable
3	19	No	Fever	No	Yes	No	Abx	Alive	Stable
4	17	No	Fever	No	No	No	-	Alive	Impaired
5	15	No	Fever	No	Yes	No	-	Alive	Stable
6	21	No	FLS	No	Yes	No	-	Alive	Stable
7	15	Yes	RF + AKI + A/A + PE	Yes	Yes	Yes	Mo-Ab, O2	Deceased	-
8	22	Yes	Pneumonia + AKI + RF	Yes	No	No	-	Deceased	-
9	24	No	Asymptomatic	No	Yes	No	-	Alive	Stable
10	22	Yes	FLS + AKI	No	Yes	Yes	-	Alive	Stable
11	10	Yes	Pneumonia	No	Yes	Yes	Abx, HCQ	Alive	Stable
12	10	No	Asymptomatic	No	No	No	-	Alive	Impaired
13	29	No	FLS	No	Yes	Yes	-	Alive	Stable
14	29	No	Para-flu syndrome	No	Yes	Yes	-	Alive	Stable
15	11	No	Asymptomatic	Yes	No	No	-	Alive	Impaired
16	27	No	FLS	No	No	Yes	Mo-Ab, Abx	Alive	Stable
17	5	No	FLS	No	Yes	No	Abx, HCQ	Alive	Stable
18	25	No	FLS	No	Yes	Yes	-	Alive	Stable
19	10	Yes	Pneumonia + RF	Yes	Yes	Yes	-	Deceased	-
20	10	Yes	Pneumonia	No	No	Yes	O2	Alive	Impaired
21	8	No	Asymptomatic	No	Yes	No	-	Alive	Stable
22	19	No	FLS	No	Yes	No	-	Alive	Stable
23	18	No	Para-flu syndrome + A/A	No	Yes	Yes	-	Alive	Stable
24	5	Yes	Pneumonia	No	Yes	Yes	-	Alive	Impaired
25	16	No	FLS + A/A	No	Yes	No	-	Alive	Stable
26	7	No	FLS	No	Yes	Yes	-	Alive	Stable
27	2	No	Asymptomatic	No	Yes	No	-	Alive	Stable
28	10	No	Asymptomatic	No	Yes	No	Abx	Alive	Stable
29	1	Yes	FLS + AKI	Yes	Yes	No	-	Alive	Impaired
30	15	No	FLS	No	Yes	No	Mo-Ab	Alive	Stable
31	1	Yes	Asymptomatic	No	Yes	No	-	Alive	Stable
32	1	Yes	Pneumonia + AKI+ RF	No	Yes	No	O2	Alive	Stable
33	11	No	FLS	No	Yes	Yes	-	Alive	Impaired
34	10	Yes	FLS	No	Yes	Yes	-	Alive	Stable
35	1	No	FLS	No	Yes	Yes	-	Alive	Stable
36	8	No	FLS	No	Yes	Yes	-	Alive	Stable
37	6	No	FLS	No	Yes	Yes	-	Alive	Stable
38	7	Yes	Pneumonia + A/A	No	Yes	Yes	O2, LMWH	Alive	Stable
39	5	Yes	FLS + AKI + A/A	No	Yes	No	-	Alive	Impaired
40	1	No	Para-flu syndrome	No	Yes	Yes	-	Alive	Stable
41	1	No	Asymptomatic	No	Yes	Yes	-	Alive	Stable

Abbreviations: ↓, reduction; ↑, increase; A/A, ageusia/anosmia; Abx, antibiotics; AKI, acute kidney injury; CNI, calcineurin inhibitors; FLS, flu-like syndrome; HCQ, hydroxychloroquine; LMWH, low-molecular-weight heparin; MMF, mycophenolate mofetil; Mo-Ab, monoclonal antibodies; MP, methylprednisolone; O2, oxygen therapy; PE, pulmonary embolism; RF, respiratory failure.

## Data Availability

The anonymised data presented in this study are available on request from the corresponding author. The dataset is not publicly available due to institutional limitations policy (authorised research purposes only).

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
