# Peer review of "Outcomes of Patients Receiving a Kidney Transplant or Remaining on the Transplant Waiting List at the Epicentre of the COVID-19 Pandemic in Europe: An Observational Comparative Study"

_pathogens, 2022, doi:10.3390/pathogens11101144_

Round 1
Reviewer 1 Report
This paper has been overtaken by events. Transplants in many (but not all) centers declined during Covid, but then picked up--the authors should mention this. The quality of the English is good, but not quite up to standard, and the authors should engage a good native English writer to improve the quality of the English. There may be more data than is strictly necessary.
Author Response
We are grateful to Reviewer 1 for the feedbacks and suggestions.
Q1) Transplants in many (but not all) centers declined during Covid, but then picked up--the authors should mention this.
A1) As correctly pointed out by Reviewer 1, in the Introuction and Discussion sections, we mentioned that KT activity declined in many, but not all transplant centers during the pandemic (please, see specific comments).
Q2) The quality of the English is good, but not quite up to standard, and the authors should engage a good native English writer to improve the quality of the English.
A2) As requested, the entire manuscrit has been re-edited by a native English writer.
Q3) There may be more data than is strictly necessary.
A3) As suggested, we reduced the information contained in the Result section (main text and Tables). However, extra data were included to address Reviewer 2 comments.
Reviewer 2 Report
This is an interesting review of a single center study from Perego and colleagues focusing on the performance of the kidney transplant program before, and during the pandemic. The study specifically focused on safety and short term outcomes.
For the most part, the study achieved its aims. There are a few things that need improvement:
1. Some of the numbers don't add up (lines 194-196). There were 245 kidney transplant recipients included in the study (but adds up to only 243)
2. The authors have to do a better job of explaining the differences in creatinine. Simply stating that less than adequate follow up is not enough. As we know, creatine at 1 year will determine long term graft outcomes (Post-transplant renal function in the first year predicts long term kidney transplant survival. Hariharan S et. al. Kidney Int 2002, July 62 (1) 311-318).
What I would like them to address is:
1. Quality of donor kidneys (prior to and during the pandemic), some objective data including KDPI etc.
2. Objective data on recipients (EPTS score etc.)
3. Biopsy data if available (very important) to see if there is a difference between the two time groups.
4. While the tacrolimus levels did not reach significance, there tended to be a lower level (table 4) in the COV KTR group.
Author Response
We are grateful to Reviewer 2 for the feedbacks and comments.
Q1) Some of the numbers don't add up (lines 194-196). There were 245 kidney transplant recipients included in the study (but adds up to only 243).
A1) As correctly pointed out by Reviewer 2, we changed the numbers.
Q2) The authors have to do a better job of explaining the differences in creatinine. Simply stating that less than adequate follow up is not enough. As we know, creatine at 1 year will determine long term graft outcomes (Post-transplant renal function in the first year predicts long term kidney transplant survival. Hariharan S et. al. Kidney Int 2002, July 62 (1) 311-318).
A2) As requested, we further explored the finding related to 1-year allograft function. Donor, recipient, and allograft characteristics were comparable. Similarly, we could not find differences in KDPI, KDRI, Karpinski score, EPTS, and Italian recipient case mix index, possibly explaining the difference in serum creatinine. CIT before and during the pandemic were equivalent. Although not statistically significant, we noticed a increase in DGF and post-operative surgical complications in the COVID group. Also, the occurrence of post-transplant SARS-CoV-2 infection could have played a role. In this regard, please, consider the following paragraph in the Discussion section: "One-year serum creatinine, a surrogate marker of long-term allograft function and survival, was slightly higher among patients transplanted during the COVID period than controls (1.30 vs 1.46 mg/dL). This finding is difficult to explain as donor-, transplant-, and recipient-related characteristics, as well as the KDPI, KDRI, pre-implantation Karpinski score, EPTS, and national recipient case mix index of pre-COVID and COVID KT patients were overall similar. Other variables, more difficult to measure, may have contributed such as the reduction in outpatient follow up visits, a less timely diagnosis of adverse events, and the resulting delay in care. This was particularly true during the very early stages of the pandemic, when most patients hesitated in attending hospital care due to the perceived risk of contagion, and when the number of active members of the nephrology team dedicated to outpatient clinics was cutdown. As a matter of fact, many nephrologists previously involved in outpatient post-KT follow-up activities were employed in extraordinary tasks in accident and emergency departments or COVID-19 wards. Also, the occurrence of SARS-CoV-2 infection may have played a role as COVID-19 has been associated with acute kidney injury and irreversible loss of renal function in the general population and transplanted subjects. The short- and long-term effects of immunosuppression minimization in case of subclinical or overt disease should also be considered as much as the discretional use of basiliximab over rATG observed at the beginning of pandemic. In fact, both factors might have determined an increase in subclinical rejection episodes, with associated chronic allograft damage. Finally, we believe that the re-organization of the non-elective surgical activity as much as the very early post-operative care during the pandemic peak could have caused an increase in post-transplant surgical and medical complications. Undoubtedly, due to scarcity of personnel available, we often had to operate out of hours, in non-dedicated theatres, with anaesthesiologists, scrub and ward nurses lacking in specific transplant expertise.
Q3) Quality of donor kidneys (prior to and during the pandemic), some objective data including KDPI etc.
A3) As suggested, we calculated KDPI and KDRI.
Q4) Objective data on recipients (EPTS score etc.).
A4) As suggested, we calculated EPTS and Italian recipient case mic index.
Q5) Biopsy data if available (very important) to see if there is a difference between the two time groups.
A6) Unfortunately, baseline biopsy was available in a limited number of cases (n=68). Histology-related information have been included in the revised version of the manuscript.
Q7) While the tacrolimus levels did not reach significance, there tended to be a lower level (table 4) in the COV KTR group.
Q8) As detailed below, tacrolimus C0 in the COVID group was equivalent or marginally higher than controls:
Tacrolimus trough levels (ng/mL): After 1 month After 3 months After 6 months After 9 months After 12 months |
9.1 (7.7-11.2) 8.3 (7.0-9.8) 8.3 (6.8-10.0) 7.9 (6.7-9.4) 7.8 (6.5-9.2) |
9.2 (7.3-13.4) 9.2 (8.1-11.7) 8.7 (7.8-10.1) 8.1 (6.1-9.5) 7.4 (6.5-9.9) |
0.768 0.007 0.359 0.636 0.997 |
Round 2
Reviewer 2 Report
The authors have address all my questions. I still feel that biopsy data is important in the long term management. Hopefully in the future.